# Getting ViT in Shape:
# Scaling Laws for Compute-Optimal Model Design

**Ibrahim Alabdulmohsin**[⋆], **Xiaohua Zhai**[⋆], **Alexander Kolesnikov**, **Lucas Beyer**[⋆]
Google DeepMind
Zürich, Switzerland
`{ibomohsin,xzhai,akolesnikov,lbeyer}@google.com`

## Abstract

Scaling laws have been recently employed to derive compute-optimal model size (number of parameters) for a given compute duration. We advance and refine such methods to infer compute-optimal *model shapes*, such as width and depth, and successfully implement this in vision transformers. Our shape-optimized vision transformer, SoViT, achieves results competitive with models that exceed twice its size, despite being pre-trained with an equivalent amount of compute. For example, SoViT-400m/14 achieves 90.3% fine-tuning accuracy on ILSRCV2012, surpassing the much larger ViT-g/14 and approaching ViT-G/14 under identical settings, with also less than half the inference cost. We conduct a thorough evaluation across multiple tasks, such as image classification, captioning, VQA and zero-shot transfer, demonstrating the effectiveness of our model across a broad range of domains and identifying limitations. Overall, our findings challenge the prevailing approach of blindly scaling up vision models and pave a path for a more informed scaling.

## 1 Introduction

The de-facto approach for improving performance of vision and language models today is scale: large models are trained on more data for longer [64, 43, 24, 19, 80, 23, 13, 16]. Empirically, it has been observed that the benefit of scale often follows a predictable power law in which the performance $f(x)$ (e.g. error rate or log-perplexity) satisfies $f(x) \sim \beta x^{-c} + \varepsilon_\infty$ for some $\beta, c > 0$ as one varies the scaling dimension $x$ (e.g. data or model size), if the remaining dimensions are not bottlenecks [34, 39, 27, 26, 3, 1]. Here, $\varepsilon_\infty$ is the irreducible loss.

However, the simple power-law relation becomes more complicated when compute is considered. In this case, power laws are observed *only* along the compute-optimal frontier. Otherwise, scaling up the model size for a fixed compute budget can deteriorate performance (see [39, 35] and Figure 4). Since one often has a fixed compute budget in mind (e.g. available hardware and time), one should pick the model size that maximizes performance subject to the compute budget constraint, which may imply not training until convergence. Indeed, this approach was used successfully in the recent Chinchilla [35] that outperformed its predecessor Gopher [55] despite being $4\times$ smaller in size.

Unfortunately, in both [39] and [35] among others, the "size" of a model is equated with its parameter count, with no special consideration for model "shape dimensions", such as "depth" or "width". The rationale behind this choice follows from the surprising observation that the transformer shape had little impact on its scaling behavior in language modeling (LM) when performance is measured upstream (e.g. using log-perplexity) [39, 32, 33]. Nevertheless, follow-up analysis suggests that shape plays a pivotal role in other domains, such as in machine translation [47] and also in language modeling for *downstream* performance [66], with recent works even advocating for extreme aspect ratios, such as a single wide attention layer [12].

---

[⋆]Significant technical contributions.

37th Conference on Neural Information Processing Systems (NeurIPS 2023).

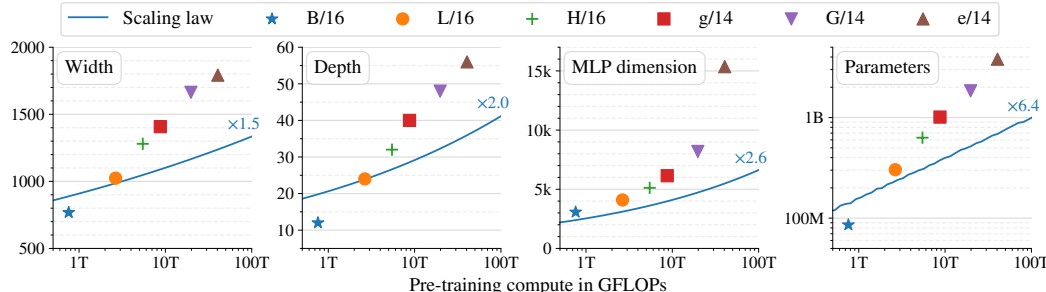

Figure 1: Predicted efficiency frontier (depth, width, MLP dimension, and parameter count) in SoViT. In large models, optimal shapes follow a similar trajectory in both image classification and multimodal tasks (see Section 4) although they can be different in small models (see Figure 3). We provide on the right (in blue) the amount of increase when compute goes from 1T to 100T GFLOPS.

In vision, in particular, much earlier works using convolutional neural networks (CNNs) pointed out that the parameter count is indeed a poor predictor of performance. For example, scaling all dimensions [64, 43, 5] in ResNets [29] is more effective than scaling a single dimension such as depth alone. In addition, scaling width [79] is often more effective than depth, especially for small models [36, 58, 75]. Hence, optimizing the "shape" of transformers seems worthwhile.

In this work, we present **SoViT**: a **s**hape-**o**ptimized **vi**sion **t**ransformer [24] that matches the performance of much larger models despite being pre-trained with equal compute. It is derived from a recipe we introduce for optimizing the shape of neural architectures, such as their depth and width. A principled approach for scaling multiple dimensions is advantageous because although one can scale dimensions via brute-force search, this requires extensive computation and often remains sub-optimal [64]. Our recipe allows us to extrapolate without having to conduct an extensive set of experiments. For example, after only 115 experiments, we identify a scaling strategy in ViT for *all* three dimensions: width (internal representation), depth, and MLP size. For comparison, [35] requires over 400 experiments to optimize a single dimension (the parameter count) alone.

One major finding is that small vision models can perform on par with larger ones with the *same compute* if we optimize their shape. In language, recent works have demonstrated the value of scaled-down architectures, such as the Chinchilla model [35] discussed earlier — a 70B parameter model that outperforms the 280B-parameter Gopher [55] and 175B-parameter GPT3 [13] — as well as LLaMA with its 13B parameter variant outperforming GPT3 on most benchmarks [69]. By introducing SoViT, we establish this phenomenon in vision as well.

Figure 1 summarizes how the various shape dimensions are scaled in SoViT (see Section 3 for derivation). The MLP dimension is scaled faster than depth, which in turn is scaled faster than width. When summarized by their parameter count (rightmost plot), compute-optimal ViTs are smaller than was previously used. With this scaling strategy, we find the shape of a ViT for the compute-equivalent of ViT-g/14 [80] pretrained on 16B JFT images [63]. We call this 2.5× smaller model SoViT-400m/14. It achieves 90.3% fine-tuning accuracy on ILSRCV2012 [22] and 82.2% zero-shot accuracy in the locked-image text tuning (LiT) setup [81]. We further evaluate SoViT-400m/14 on captioning, VQA and panoptic segmentation and highlight some results in Figure 2.

**Statement of Contribution.** In summary, our contribution is to:

- Introduce a new method for optimizing *the shape* of neural networks, such as their depth and width. Our technique expands and improves previous methods by optimizing *multiple* shape dimensions *jointly* while requiring significantly fewer experiments.

- Demonstrate the effectiveness of scaled-down architectures in vision. We optimize ViT for the compute-equivalent of ViT-g/14, leading to a smaller, faster model of equal quality.

- Present new qualitative insights for scaling vision transformers, such as on how to scale individual shape dimensions and how optimal ViT shapes vary across domains.

- Conduct extensive evaluation across tasks like image classification, image captioning, VQA, zero-shot classification and panoptic segmentation, identifying both gains and limitations.

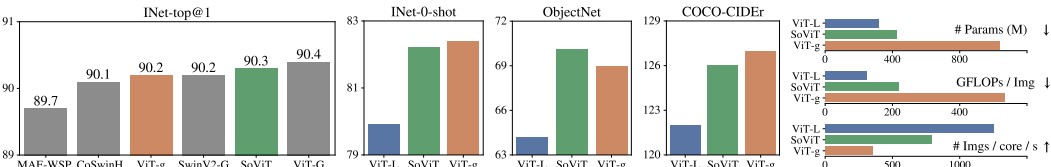

Figure 2: Optimizing for the compute-equivalent of ViT-g/14 results in the $2.5\times$ smaller SoViT-400m/14 model achieves equivalent results across a wide range of benchmarks. Our model performs exceptionally well on the competitive ImageNet (ILSRCV2012) benchmark in comparison with significantly larger models from the recent literature [61, 78, 49, 80].

## 2   Related Work

Optimizing training for compute has received a significant amount of attention in recent years, partly due to the financial and environmental costs of training large models [52, 55]. However, conflicting results are sometimes reported. For example, in language modeling, [39] argues that the model size should be scaled faster than the data size, implying it is compute optimal to "undertrain" large models. Similar conclusions are found in [47]. On the other hand, [35] argues that the model size should be scaled uniformly with the data size, and highlights that transformers were not trained long enough, leading to some recent efforts [69] "overtraining" their models instead. Our analysis for ViT in Section 4 agrees partially with the latter result.

Scaling the size of vision transformers has led to remarkable results achieving, for instance, 90.4% top-1 accuracy on ImageNet (ILSRCV2012) with 2 billion parameters [80] and 90.9% top-1 accuracy with 4 billion parameters [15]. When scaled to 22 billion parameters, ViT exhibits state-of-the-art alignment to human visual perception in terms of shape/texture bias, among other findings [21].

Despite the clear benefit of scale, there has been little investigation into optimally scaling the shape of ViTs. [66] suggest preferentially increasing depth before scaling other dimensions uniformly. For ViT, however, they only consider small ViT-S and ViT-B models and the reported accuracy improvement comes with an *increase* in FLOPs of up to $\times 4$, making it difficult to draw conclusions about the suggested shape's quality. In contrast [12] recommend scaling width over depth, but the authors do not observe any improvement when applying their strategy to ViT.

Our analysis draws inspiration from "compound scaling" in MobileNet [36] and EfficientNet [64], while differing in significant ways. EfficientNet uses an exhaustive grid search to determine the optimal architecture for a fixed increase in compute (e.g. $\times 2$). Afterwards, each dimension is scaled up by the same ratio with every subsequent increase in compute. In contrast, we expand scaling laws to simultaneously account for model size and compute beyond the efficient frontier and leverage them to derive the optimal scaling exponents for each dimension separately, as outlined in Section 3.

Throughout our analysis, we use *downstream* metrics, e.g. ImageNet 10-shot error, when measuring performance instead of upstream metrics. This follows recent reports arguing that upstream performance may not reflect downstream performance in language and vision [65, 80].

We use GFLOPs as a proxy for compute since it is hardware-agnostic and correlates well with actual wall-clock core-hours (see Figure 4). However, GFLOPs can have limitations [5, 20] and may not be a perfect predictor for the metric of interest (e.g. core hours) in all model and hardware types. Note that we focus on scaling the shape of the architecture, not on improving its training protocol, which can be similarly beneficial [5, 67, 62, 68].

## 3   Scaling Strategy

**Notation.** We begin with a formal description of the problem. We represent a neural architecture as a tuple $\mathbf{x} = (\mathbf{x}_1, \mathbf{x}_2, \ldots, \mathbf{x}_D) \in \mathbb{N}^D$ containing $D$ shape dimensions, such as width, depth and MLP size. We denote compute such as GFLOPs by $\mathbf{t}$. We designate $f : \mathbb{N}^D \times \mathbb{R}^+ \to \mathbb{R}$ a performance metric of interest, such as downstream ImageNet 10-shot error rate. Specifically, $f(\mathbf{x}, \mathbf{t})$ results from (pre-)training an architecture $\mathbf{x}$ for a fixed compute budget $\mathbf{t}$. We always assume that $f$ corresponds to a loss, meaning lower values are better.

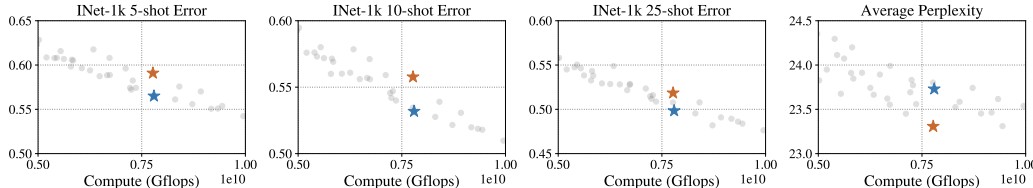

Figure 3: A grid sweep over multiple ViT shapes pretrained on 600M JFT examples highlights the important role of shape. Each dot corresponds to a model architecture pretrained on 600M examples and evaluated on a downstream metric, e.g. Imagenet-1k 5-shot in the leftmost plot. The two architectures marked in blue and red – identical in all four figures – are compute-optimal for classification and image-to-text tasks (captioning/VQA), respectively. For captioning/VQA, we average log-perplexity scores (see Section 4.2). In the leftmost three figures, using Imagenet-1k few-shot evaluation, the compute-optimal model highlighted in blue is compute-optimal in all three cases, but it is not compute-optimal for image-to-text tasks as shown in the rightmost figure. So, in *small* models, an optimal shape in one domain is not necessarily optimal in others.

The goal of optimizing shape for fixed compute $\mathbf{t}$ is to identify $\mathbf{x}^\star$ (depending on $\mathbf{t}$) such that:

$$f(\mathbf{x}^\star, \mathbf{t}) - \inf_{x \in \mathbb{N}^D} f(x, \mathbf{t}) \leq \epsilon, \tag{1}$$

for some small tolerance $\epsilon > 0$. Due to modeling assumptions, approximations, and the finite possible number of experiments conducted, we cannot hope for $\epsilon = 0$ and have to tolerate a small excess loss.

**Single Dimension.** As demonstrated in Figure 3, the shape of a pretrained vision transformer has an impact on its downstream performance. To determine an optimal shape scaling strategy, we begin by considering both compute $\mathbf{t}$ and a *single* shape dimension $\mathbf{x}_k$ for $k \in [D]$, such as depth. In prior works, optimizing a single dimension $\mathbf{x}_k$ for compute involves running a large number of experiments in order to identify the Pareto optimal frontier, from which power laws on $\mathbf{x}_k$ or $\mathbf{t}$ are derived [39, 35]. Since this is expensive, we propose the following joint functional form instead:

$$f_k(\mathbf{x}_k, \mathbf{t}) \sim \alpha_k \mathbf{x}_k^{-a_k} + (\beta_k \mathbf{x}_k^{b_k} + \xi_k) \mathbf{t}^{-c} + \varepsilon_k, \tag{2}$$

where $\alpha_k, a_k, \beta_k, b_k, c, \xi_k, \varepsilon_k > 0$. Here, $f_k$ focuses on the dimension $k$ alone and assumes that all other shape dimensions $j \neq k$ are sufficiently large such that they do not constitute a bottleneck. We also assume that data is unlimited so that there is no risk of overfitting. We estimate the parameters in (2) by minimizing the *relative* error. In (2), $a_k$ are scaling exponents when varying the corresponding shape dimension in the compute-unbounded regime, $c$ is the data scaling exponent, while $b_k$ relates to the impact of the model shape on compute.

Our argument for this particular functional form is six-fold:

I. If compute is unbounded, we recover the familiar power law relation on model size $f_k(\mathbf{x}_k) \sim \alpha_k \mathbf{x}_k^{-a_k} + \varepsilon_k$ [34, 2, 38, 39]. In addition, increasing the model size $x_k$ while keep the data size fixed does not imply that $f_k(\mathbf{x}_k, \mathbf{t}) \to \varepsilon_k$ because $\mathbf{x}_k^b$ can increase faster than $\mathbf{t}^c$ in (2).

II. For any *fixed* model size, the relation above reduces to the power law $f_k(\mathbf{t}) \sim A\mathbf{t}^{-c} + B$, where $A = \beta_k \mathbf{x}_k^{b_k} + \xi_k$ and $B = \alpha_k \mathbf{x}_k^{-a_k} + \varepsilon_k$. Since the model size is fixed, $\mathbf{t}$ is proportional to the size of the data. Such data scaling laws have been demonstrated extensively in various domains [1–3, 27, 34, 39, 59, 80].

III. For fixed compute, the relation w.r.t. $\mathbf{x}_k$ is non-monotone, quasiconvex (see Appendix A), in agreement with empirical measurements [39, 35]. See IsoFlop curves in Figure 4.

IV. Arguments for power law behavior using space partitioning suggest that the exponent $c$ is independent of the shape dimension. In particular, $c = \Theta(1/d)$, where $d$ is the intrinsic dimension of the data manifold [2, 38, 59]. From this, we conclude that assuming the functional form in (2) for every shape dimension *separately* cannot lead to any contradictions since this assumption is satisfied by the decomposable loss:

$$f(\mathbf{x}, \mathbf{t}) = \sum_k \alpha_k \mathbf{x}_k^{-a_k} + \sum_k \beta_k \mathbf{x}_k^{b_k} \mathbf{t}^{-c} + \xi \mathbf{t}^{-c} + \varepsilon_\infty, \tag{3}$$

for some constants $\xi, \varepsilon_\infty > 0$.

V. When optimizing the shape dimension $\mathbf{x}_k$ for fixed compute $\mathbf{t}$, the optimal value $\mathbf{x}_k^\star$ is:

$$\mathbf{x}_k^\star = \left(\frac{\alpha_k\, a_k\, \mathbf{t}^c}{\beta_k b_k}\right)^{\frac{1}{b_k+a_k}} = O\left(\mathbf{t}^{s_k}\right), \quad \text{where: } s_k = \frac{c}{b_k + a_k}. \tag{4}$$

Recall that the scaling exponent $s_k$ in (4) is positive because $a_k, b_k, c > 0$. Using the relation (4), we rearrange the terms in Eq. (2), and obtain the scaling law for model performance along the compute-optimal frontier (Appendix A):

$$f_k(\mathbf{x}_k, t) = F\mathbf{x}_k^{-a_k} + G\mathbf{t}^{-c} + \varepsilon_k, \qquad \text{(in the compute-optimal frontier)} \tag{5}$$

for some constants $F$ and $G$, which is a sum of power law terms involving the model size and compute. Indeed, this decomposition has been demonstrated to hold within the compute-optimal frontier by [39] and [35].

VI. Eq. (2) fits empirical measurements and extrapolates accurately as well, see Figure 4.

**Multiple Dimensions.** Next, we expand upon the previous approach by incorporating multiple dimensions. To reiterate, our method involves both a functional form (2) and a novel procedure. Our procedure significantly decreases the number of large-scale experiments required to identify compute-optimal architectures, by an order of magnitude compared to prior work [35].

*Star Sweep* – Conducting a brute-force grid search to estimate scaling parameters across all dimensions is expensive, since it requires $O(2^D)$ experiments to cover the search space. Instead, we demonstrate that a "star sweep" is sufficient: (1) starting from a *large* model $\mathbf{x}^{(c)}$ (the star center), we vary a single dimension $k \in [D]$ at a time in an exponentially-spaced grid, such that all values are much smaller than $\mathbf{x}_k^{(c)}$. In our experiments, for instance, we optimize three shape parameters: `width`, `depth`, and `MLP dim` (see Section 4 for a brief definition of each dimension). Our star center is $\mathbf{x}^{(c)} = (1968, 40, 6144)$; i.e. has `width` 1968, `depth` 40, and `MLP dim` 6144. When varying `MLP dim` in the star sweep, we use the grid $(1088, 1360, 1728, 2160, 2592, 3072)$, corresponding to about 20% increase in each step, while fixing `width` to 1968 and `depth` to 40. We do this to ensure that other dimensions do not form a bottleneck when estimating the parameters in (2). This gives us the scaling exponents $s_k$ in (4).

*Grid Sweep* – The second stage is a grid sweep for *small* models trained for *short* compute. Depending on the number of shape dimensions involved, the cost of running this grid sweep can be negligible. Its goal is to identify a single architecture $\mathbf{x}^{(0)}$ that lies in the Pareto optimal frontier for small compute as illustrated in Figure 3. This is important since a suboptimal $\mathbf{x}^{(0)}$ can significantly skew results [5]. Our grid sweep identifies $\mathbf{x}^{(0)}$ to be $(608, 10, 928)$, the blue star in Figure 3. The advantage of this step is to absorb the leading coefficients in $\mathbf{x}_k^\star = O(\mathbf{t}^{s_k})$ in (4) so that the star sweep focuses on estimating the *exponents* $s_k$ alone. We demonstrate in Figure 5 that the scaling exponents $s_k$ are robust to the choice of the evaluation metric $f$. In Appendix B.3, we discuss important considerations that were taken into account during this analysis.

**Scaling.** Finally, we scale all dimensions jointly. Starting from the small compute-optimal architecture $\mathbf{x}^{(0)}$ and the amount of compute $\mathbf{t}^{(0)}$ it is optimal for, suppose we increase compute by a factor $\tau > 1$ (i.e. the new compute is $\tau\, \mathbf{t}^{(0)}$). By treating this increment $\tau$ as a *sequence* of $D$ smaller increments of size $\tau^{w_k}$ each with $\sum_k w_k = 1$, an increase in compute by a factor of $\tau$ is accompanied by an increase in every shape dimension $k$ by a factor of $\tau^{w_k}$, respectively. In this work, the adopt the simplest strategy of setting $w_k = 1/D$, but acknowledge that more sophisticated approaches might lead to better results.

## 4 Shape-optimized ViT

We implement the scaling strategy in Section 3 in vision transformers [24] pretrained on JFT-3B, a proprietary dataset with about 30k classes and around 3 billion examples [80], using the Adam optimizer [41]. As mentioned in Section 3, we focus on optimizing three shape dimensions: `width` (size of internal representation), `depth` (number of encoder blocks) and `MLP dim` (hidden dimension). Following [43, 24, 80], we remove near-duplicate examples between upstream JFT-3B data and all the downstream train and test sets. Appendix B contains the full set of hyper-parameters used in the

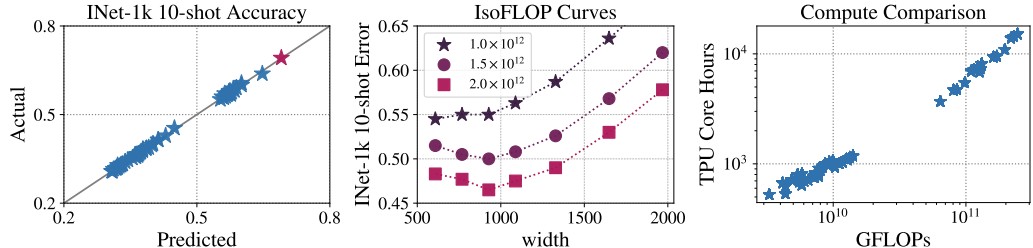

Figure 4: LEFT: Comparison between ILSRCV2012 (denoted INet-1k) 10-shot error rate predicted by Eq. (2) and actual. The value marked in **violet** corresponds to the star center $\mathbf{x}^{(c)}$ that is never used when estimating scaling parameters. Eq. (2) is consistent with empirical measurements and extrapolates accurately. MIDDLE: IsoFlop curves in ViT as one varies the width dimension. RIGHT: GFLOPs is well-correlated with actual TPU core hours across models (correlation coefficient $\sim 0.99$).

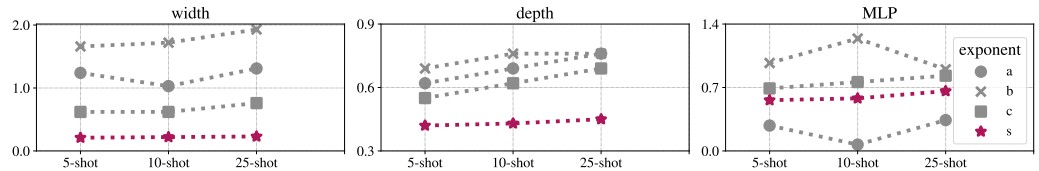

Figure 5: A plot of the estimated values of the exponents in (2) for different evaluation metrics $f$. The scaling exponent $s_k$ tends to be less sensitive to the choice of metric than other exponents. Moreover, the data scaling exponent $c$ is approximately $c \approx 0.65 \pm .06$, independently of the choice of the shape dimension, in agreement with what would be expected using space partitioning arguments [2, 38, 59].

experiments, including full details about the star and grid sweeps described in Section 3. We fix the patch size in our analysis to $14 \times 14$, but study "flexifying" to arbitrary sequence lengths following [7] in Section 5.5.

As an evaluation metric $f$, we consider two domains: (1) image classification, with ImageNet linear 10-shot error rate as the metric, and (2) image-to-text LiT-decoding following [8]. In the latter case, the evaluation metric $f$ is an average of four perplexity scores: COCO captioning, optical character recognition (OCR), and question answering (VQAv2 and GQA). Refer to [8] for details about the LiT-decoder setup. By considering such distinct domains, our goal is to identify similarities and differences (if any) in how to optimally scale the shape of vision transformers (ViT).

### 4.1 Image Classification

We use the aforementioned star center $\mathbf{x}^{(c)} = (1968, 40, 6144)$ as our starting point. To estimate the scaling exponents $s_k$ in (4) for each dimension separately, we vary `width` in the grid (608, 768, 928, 1088, 1328, 1648), `depth` in the grid (8, 10, 12, 16, 20, 24), and `MLP dim` in the grid (1088, 1360, 1728, 2160, 2592, 3072). As discussed in Section 3, we use an exponential spacing with all values being much smaller than in the star center $\mathbf{x}^{(c)}$. Following [24], we evaluate quality using few-shot linear transfer by using pre-trained models to extract features and fitting a linear regression head mapping them to the one-hot encoding of the target labels.

The individual scaling exponents we find are $s_{\text{depth}} \approx 0.45$, $s_{\text{width}} \approx 0.22$, and $s_{\text{MLP}} \approx 0.6$. Importantly, these exponents are quite robust to the choice of the metric. As shown in Figure 5, changing the metric from ImageNet 10-shot to either 5-shot or 25-shot can change the best-fit estimate of the other exponents $a_k, b_k, c_k$ in (2) but the scaling exponent $s_k$ is relatively unchanged, since it is formed as a *ratio* over other exponents. In addition, the data scaling exponent $c$ appears to be independent of the choice of the shape dimension. As mentioned earlier, this is consistent with space partitioning arguments for power law scaling [2, 38, 59].

The estimated scaling exponents $s_k$ point to the following picture:

    I. MLP dimension should be scaled faster than depth, and depth faster than width.

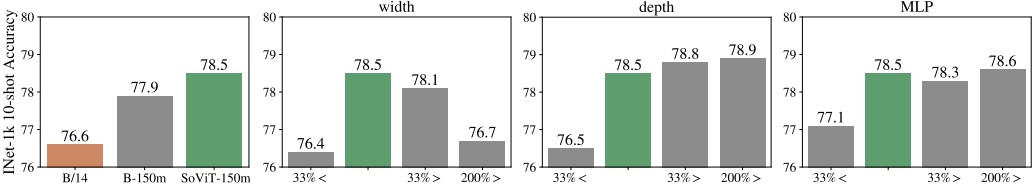

Figure 6: LEFT: Optimizing ViT shape for the compute-equivalent of ViT-B/14 results in SoViT-150m/14, which improves performance significantly. See Section 4.1. CENTER & RIGHT: Impact of deviating from the optimal shape in SoViT-150m/14 (in green) while keeping compute fixed by changing the training duration such that the total FLOPs is the same in all models.

II. The size of ViT, as quantified by its parameter count, is scaled more slowly than the allocated compute. More precisely, for every increment in compute by a factor of 10, the parameter count of the optimized model shape increases by a factor of $\approx 2.5$.

III. As demonstrated in Figure 1, small ViT models can match the performance of much larger ones when their shape and training duration are jointly optimized for the available compute.

We validate these predictions by optimizing the shape of ViT for the compute-equivalent of ViT-g/14 when the latter is pretrained on 16 billion JFT-3B examples as done in [80]. The resulting model, SoViT-400m/14, is significantly smaller and faster, yet equally competitive. It has a `width` of 1152, `depth` 27, and `MLP dim` 4304. Fine-tuning it on ImageNet results in a 90.3% top-1 accuracy, see Figure 2. Section 5 presents various other evaluations.

In Figure 6, we also optimize the shape of ViT for the compute-equivalent of ViT-B/14 pretrained on 4 billion examples of JFT-3B using Imagenet 10-shot error rate as an evaluation metric, resulting in SoViT-150m/14. It has a `width` of 880, `depth` 18, and `MLP dim` 2320. As shown in Figure 6, optimizing the shape of ViT leads to a significant improvement in performance, from 76.6% in ViT-B/14 to 78.5% in SoViT-150m/14 when both are trained for the same amount of compute. We also vary the optimized shape by decreasing/increasing one dimension at a time and retraining the corresponding model while keeping the total compute fixed. As shown in Figure 6, small deviations from the predicted optimal shape can lead to a notable drop in performance, especially for width since it has the smallest scaling exponent (see Figure 5). We also include in Figure 6 (LEFT) a comparison with a model, denoted B-150m, which has the same *shape* as ViT-B/14 but the same *size* as SoViT-150m/14. This confirms that while optimizing the model size improves performance, optimizing the shape improves it even further.

Importantly, the model shapes in Figure 6 bear no resemblance to those observed during the star or grid sweeps. To recall, the star sweep is centered around an architecture $\mathbf{x}^{(c)}$ whose shape dimensions are significantly larger than in ViT-B/14, whereas the grid sweep pretrains models that are substantially smaller and for only 600M examples. The ability of our strategy to accurately identify a near-optimal model shape within this context underscores its robust extrapolation capability.

## 4.2 Multitask Decoder

Besides image classification, there has been a significant interest in multimodal applications, mostly fueled by the convergence across language and vision on the transformer architecture [72, 24]. In particular, an encoder-decoder transformer with an autoregressive decoder is a popular choice because it allows reusing pretrained image encoders. We repeat the analysis conducted in Section 4.1 to optimize the shape of the image encoder, while fixing the decoder architecture to two layers as was used in [8]. Further details are provided in Appendix C. As an evaluation metric $f$, we use the average of four perplexity scores: COCO captioning [48, 14], OCR [50], VQAv2 [28] and GQA [37], without normalization since they share a similar scale. For the learning rate and weight decay hyper-parameters, we conduct a sweep where we vary the learning rate in $\{10^{-3}, 3 \times 10^{-4}, 10^{-4}\}$ and the weight decay in $\{3 \times 10^{-4}, 10^{-4}, 3 \times 10^{-5}\}$. We pick the largest learning rate and the corresponding weight decay that result in a stable training run (i.e. smooth training loss curve and gradient norms) for both the largest and smallest image encoder architectures. From this, a learning rate of $3 \times 10^{-4}$ and a weight decay of $10^{-4}$ are selected.

Table 1: ImageNet fine-tuning. The top shows models trained in the same controlled setting, and the bottom a representative set of large well-performing models. SoViT compares favorably. Contrary to common practice, we use a held-out 2% of Train to select hyper-parameters. Selecting them on Val would increase all scores. FLOPs according to XLA; PyTorch reports MACs.

| Model | Pretraining | Size | | | ImageNet variant | | |
|---|---|---|---|---|---|---|---|
| | | Input | Params | FLOPs | Val [57] | ReaL [6] | v2 [56] |
| SoViT-400m/14 | JFT-3B | $224^2$ | 428 M | 221 G | 88.9 | 90.3 | 80.7 |
| ViT-L/16 [80] | JFT-3B | $384^2$ | 303 M | 383 G | 88.5 | 90.4 | 80.4 |
| SoViT-400m/14 | JFT-3B | $384^2$ | 428 M | 672 G | 90.0 | 90.9 | 83.2 |
| ViT-g/14 [80] | JFT-3B | $518^2$ | 1011 M | 3208 G | 90.2 | 90.9 | - |
| SoViT-400m/14 | JFT-3B | $518^2$ | 428 M | 1374 G | 90.3 | 91.0 | 83.4 |
| ViT-G/14 [80] | JFT-3B | $518^2$ | 1882 M | 5668 G | 90.4 | 90.8 | 83.3 |
| SwinV2-G [49] | IN-21k + 70M | $640^2$ | 3000 M | - | 90.2 | - | 84.0 |
| CoAtNet-6 [19] | JFT-3B | $512^2$ | 1470 M | 1521 G | 90.4 | - | - |
| MAE→WSP [61] | IG-3B | $518^2$ | 1890 M | 5679 G | 89.7 | 90.9 | 83.0 |
| CoCa [77] | JFT-3B + ALIGN-1.8B | $576^2$ | 2100 M | - | 91.0 | - | - |

Using this analysis, the derived scaling exponents are approximately $0.25, 0.49$ and $0.62$ for `width`, `depth` and `MLP size`, respectively. Hence, whereas the optimal shape dimensions in small architectures can be quite different between image classification and multitask decoding, as shown in Figure 3, the scaling exponents are nearly identical, so the same scaling recipe is used in both domains.

## 5  Evaluations

**Overview.** We now evaluate SoViT-400M in various contexts to verify whether it broadly matches ViT-g/14's performance, or only in the ILSRCV2012 10-shot metric it was optimized for. The settings we cover are few-shot, frozen linear probes on ImageNet, zero-shot transfer, image-language multitasking including captioning, OCR, and question answering, as well as panoptic segmentation. In each of these settings, we compare SoViT-400m/14 to ViT-L/16 and a ViT-g/14, all trained on the

**Compute.** Experiments are executed on Tensor Processing Units (TPU). SoViT-400m/14 is pretrained on 40 billion examples, which amounts to 9T GFLOPs and 230K TPUv3 core-hours. ViT-g/14 was pretrained on 16 billion examples, corresponding to 9T GFLOPs and 210K TPUv3 core-hours.

### 5.1  Image Classification

We verify classification performance in three common and widely useful setups: full fine-tuning, linear probes on the frozen model, and few-shot linear classification.

**Fine-tuning on ImageNet.** Pre-trained image encoders are most commonly [18] evaluated by fine-tuning them on the ILSVRC2012 classification task. The detailed fine-tuning settings are provided in Appendix E. One important aspect is to increase image resolution [70] as a way of further increasing the capacity of the pre-trained model during fine-tuning [43]. Table 1 shows the performance of SoViT-400m/14 in comparison with ViT-L/16, ViT-g/14 fine-tuned at various resolutions, along with a few more representative models from the literature. The results confirm that SoViT-400m/14 achieves the goal of matching ViT-g/14 while being significantly smaller.

**Linear probing on ImageNet.** The quality of the pretrained representation learned by the model is often more directly assessed by performing *linear probes*, meaning learning a linear classifier on top of unmodified, frozen output features from the model. We present results of this evaluation on the full ImageNet-1k [57] dataset in Table 2, including robustness evaluations of the learned

Table 2: Linear ILSVRC2012 probes.

| | Val | ReaL | v2 | -R | -A | Obj |
|---|---|---|---|---|---|---|
| L/16 | 86.7 | 90.0 | 78.5 | 88.9 | 67.8 | 63.5 |
| SoViT | 88.2 | **90.3** | 80.6 | 89.0 | 76.4 | **68.7** |
| g/14 | **88.4** | 90.2 | **80.8** | **90.3** | **76.6** | 67.7 |

Table 3: SoViT-400m/14 performs competitively with ViT-g/14 in 10-shot classification.

| | INet [22] | CIFAR100 [46] | Pets [51] | Birds [74] | Caltech [25] | Cars [45] | Colorectal [40] | DTD [17] | UC [76] |
|---|---|---|---|---|---|---|---|---|---|
| ViT-L/16 | 81.5 | 82.2 | 97.0 | 97.1 | 89.9 | 93.8 | 79.4 | 72.0 | 96.3 |
| SoViT-400m/14 | **84.1** | 86.7 | **97.6** | **88.8** | **91.3** | 93.6 | **81.5** | 72.5 | 97.7 |
| ViT-g/14 | 84.0 | **87.2** | 97.4 | 88.5 | 89.3 | **93.9** | 78.9 | **74.1** | **98.2** |

probe according to ReaL [6], ImageNet-v2 [56], ImageNet-Renditions [30], ImageNet-Adversarial [31], and Object-Net [4] testsets. SoViT-400m/14 is generally on par with ViT-g/14 despite its smaller output width.

**Broad few-shot linear transfer.** We follow [24, 80] and evaluate a closed-form linear regression probe for 10-shot classification across a wide range of classification tasks in Table 3. Again, SoViT-400m/14 performs on-par with ViT-g/14 across the board.

## 5.2 Contrastive image-text tuning

Next, we follow the locked-image text tuning (LiT) recipe [81] on the WebLI dataset [15] to add zero-shot classification abilities to the pre-trained ViT-L/16, SoViT-400m/14 and ViT-g/14 image encoders. In this setup, a new text encoder is trained using the contrastive image-text matching objective [54]. See Appendix D for details. Table 4 (second column) shows that SoViT-400m/14 is competitive with ViT-g/14, and substantially better than ViT-L/16.

## 5.3 Multitask Decoding

We also evaluate the three pretrained ViT models in multitask decoding as described in Section 4.2, where we follow the setup studied in [8]. We fix the decoder architecture to two layers since it was found to perform well [8]. For evaluation, we report COCO CIDEr [48, 14, 73], OCR [50], VQAv2 [28] and GQA [37] accuracy and log-perplexity. In brief, the CIDEr score measures the similarity between a generated caption and reference captions, considering $n$-gram statistics, OCR evaluates optical character recognition, whereas both VQAv2 and GQA are question-answering evaluations. Results are summarized in Table 4. SoViT-400M performs on par with ViT-g/14.

## 5.4 Panoptic Segmentation

Additionally, we evaluate SoViT-400m/14 on panoptic segmentation [42], which is a challenging dense scene understating task by closely following the setup in UViM [44]. At a high level, UViM panoptic segmentation model consists of a visual image encoder and a decoder which maps the image representation to an intermediate code. The code is later decoded to the panoptic segmentation mask using a fixed VQVAE [71] model, which was pretrained on panoptic masks [44]. In our experiments we initialize UViM's image encoder with ViT-L/16, SoViT-400m/14 and ViT-g/14.

Following [44], we train the UViM model using the COCO panoptic dataset (with $512 \times 512$ input resolution) and report the PQ metric. We achieve 43.5, 43.7 and 44.8 PQ points for ViT-L/16, SoViT-400m/14 and ViT-g/14 respectively. Our results indicate that dense segmentation tasks can be a limitation of the proposed optimal model shape, and a different model shape might be derived in this domain. We leave this investigation for future work.

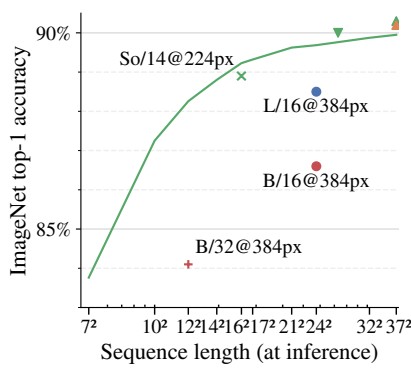

## 5.5 Flexifying SoViT-400M

Finally, since we do not include the patch size (sequence length) as part of the shape optimization, we verify that this is not a limitation by *flexifying* [7] SoViT-400m/14 on ILSVRC2012 for 300 epochs. The performance of the resulting FlexiSoViT-400m is shown in Fig 7 as green

Figure 7: Flexification of SoViT-400m/14 (abbr. So/14). See Section 5.5.

Table 4: Summary of multitask decoding and zero-shot transfer results, see Sections 5.2 & 5.3.

| Model | ImgNet | OCR-VQA [50] | | GQA [37] | | VQAv2 [28] | | COCO Capt. [14] | |
|---|---|---|---|---|---|---|---|---|---|
| | Zero-shot | Acc [%] | Log-PPL | Acc [%] | Log-PPL | Acc [%] | Log-PPL | CIDEr | Log-PPL |
| ViT-L/16 | 79.9 | 48.3 | 17.9 | 55.3 | 24.9 | 66.4 | 20.9 | 120 | 28.7 |
| SoViT-400M | 82.2 | **52.9** | **15.3** | 56.0 | 23.9 | 67.7 | 20.9 | **125** | **28.1** |
| ViT-g/14 | **82.4** | 52.5 | 15.9 | **58.0** | **22.5** | **68.8** | **21.5** | 126 | 27.9 |

curve when varying the patch-size at inference time. A
few reference ViT models from Table 1 and [80] are added, confirming that SoViT-400m maintains a
clear advantage. It is worth noting that flexifying does not rule out that other patch sizes could be
compute-optimal. It merely demonstrates that SoViT-400M continues to perform quite well for other
patch sizes when it is flexified.

# 6 Conclusion

In conclusion, we introduce an efficient method for optimizing the shape of neural architectures and
successfully apply it to vision transformers. Our analysis demonstrates that smaller models, trained
at their optimal architecture shape for the right amount of compute, can match much larger models.

## Acknowledgments and Disclosure of Funding

We thank Mostafa Dehghani, Andreas Steiner, Daniel Keysers, Neil Houlsby, Sam Smith, David
Schneider-Joseph, Rodolphe Jenatton and the anonymous reviewers for their valuable feedback and
discussions. We also thank the Google DeepMind unit at large for providing a supportive research
environment. We use the `big_vision` codebase [10, 9] for conducting experiments in this project.

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

# A  Scaling Laws Analysis

In this appendix, we present proofs of two claims in the paper. First, we show that (2) is quasiconvex on its first argument $\mathbf{x}_k$. Second, we derive (5).

## A.1  Quasiconvexity Proof

We assume throughout the proof that $a_k, b_k$ are strictly positive, otherwise $f_k(\mathbf{x}_k, \mathbf{t})$ is a monotone function on its first argument and the statement holds trivially.

To establish the quasiconvexity of $f_k(\mathbf{x}_k, \mathbf{t})$ in (2), we observe that:

$$\frac{\partial f_k}{\partial \mathbf{x}_k} = -\alpha_k a_k \mathbf{x}_k^{-(1+a_k)} + \beta_k b_k \mathbf{t}^{-c} \mathbf{x}_k^{b_k - 1} \doteq -A\mathbf{x}_k^{-(1+a_k)} + B\mathbf{x}_k^{b_k - 1}.$$

Setting the derivative to zero gives the *unique* solution in $\mathbb{R}^+$:

$$\hat{\mathbf{x}} = \left( \frac{A}{B} \right)^{\frac{1}{a_k + b_k}}.$$

At the limit $\mathbf{x}_k \to \infty$, the term involving $\mathbf{x}_k^{-a_k}$ vanishes and we have the asymptotic relation:

$$f_k(\mathbf{x}_k, \mathbf{t}) \sim \beta_k \mathbf{t}^{-c} \mathbf{x}_k^{b_k},$$

which is an increasing function since $b_k > 0$. Since $\hat{x}$ is the only point in $\mathbb{R}^+$ where $\partial f_k / \partial \mathbf{x}_k = 0$, we conclude that $f(\mathbf{x}_k, \mathbf{t})$ is monotone increasing for all $\mathbf{x}_k \geq \hat{x}$.

Similarly, when $\mathbf{x}_k \to 0^+$, we have:

$$f_k(\mathbf{x}_k, \mathbf{t}) \sim \alpha_k \mathbf{x}_k^{-a_k},$$

which is monotone decreasing. Therefore, $f'(\mathbf{x}_k, \mathbf{t}) \leq 0$ for all $\mathbf{x}_k \leq \hat{x}$. Combining both results implies that $f_k(x, \mathbf{t})$ is monotone decreasing in the domain $x \in (0, \hat{x})$ and is monotone increasing in the domain $x \in (\hat{x}, \infty)$.

A function $f(y)$ is said to be quasi-convex if for any $y_1$ and $y_2$ in its domain and any $\lambda \in [0, 1]$, one has [11]:

$$f(\lambda y_1 + (1 - \lambda) y_2) \leq \max \{ f(y_1), f(y_2) \}. \tag{6}$$

Suppose for the purpose of obtaining a contradiction that $f_k(\mathbf{x}_k, \mathbf{t})$ is not quasiconvex on its first argument. Then, there exists two points $y_1, y_2 \in \mathbb{R}^+$ and $\lambda \in [0, 1]$ such that the above condition is violated. Let $\hat{y} = \lambda y_1 + (1 - \lambda) y_2$. But, then, by the mean-value theorem, there must exist two points $c_1 \in [y_1, \hat{y}]$ and $c_2 \in [\hat{y}, y_2]$ where:

$$f_k'(c_1) = \frac{f(\hat{y}) - f(y_1)}{\hat{y} - y_1} \geq 0$$

$$f_k'(c_2) = \frac{f(y_2) - f(\hat{y})}{y_2 - \hat{y}} \leq 0,$$

with $c_2 > c_1$. This implies that $c_1 \geq \hat{x}$ and $c_2 \leq \hat{x}$, which is a contradiction. Therefore, $f_k(\mathbf{x}_k, \mathbf{t})$ is quasi-convex on its first argument.

## A.2  Derivation of (5)

Rearranging the expression in (4), we have:

$$\left( \frac{\beta_k b_k}{\alpha_k a_k} \right) (\mathbf{x}_k^{\star})^{b_k + a_k} = \mathbf{t}^c$$

From this and (2), we obtain:

$$f_k(\mathbf{x}_k^{\star}, \mathbf{t}) = \alpha_k (\mathbf{x}_k^{\star})^{-a_k} + \beta_k (\mathbf{x}_k^{\star})^{b_k} \left( \frac{\alpha_k a_k}{\beta_k b_k (\mathbf{x}_k^{\star})^{b_k + a_k}} \right) + \xi_k \mathbf{t}^{-c} + \varepsilon_k,$$

where we plugged in the last expression. Simplifying yields (5) for some constants $F, G \geq 0$.

# B  Shape Optimization

## B.1  Hyper-parameters

Table 5: Common hyper-parameters settings for both star and grid sweeps.

| | |
|---|---|
| Image Resolution | $224 \times 224$ |
| Batch size | 128 |
| Preprocessing | Rescale(-1, 1) |
| Augmentation | InceptionCrop, Left-Right Flip |
| Optimizer | AdaFactor [60] |
| Gradient Clipping | 1.0 |
| Learning Rate | 8e-4 |
| Label Smoothing | 0 |
| Weight Decay | $0.03 \times 8\text{e-}4$ |
| Schedule | Reverse SQRT, 10K Warmup steps, 50K Cooldown steps |

Table 5 provides the set of hyperparameters used in the star and grid sweeps. We use a small batch size of 128 here in order to train multiple models in parallel on small hardware topologies.

## B.2  Star Sweep

In the star sweep, we use the center $\mathbf{x}^{(c)} = (1968, 40, 6144)$ as our starting point. To estimate the scaling exponents $s_k$ in (4) for each dimension separately, we vary `width` in the grid (608, 768, 928, 1088, 1328, 1648), `depth` in the grid (8, 10, 12, 16, 20, 24), and `MLP dim` in the grid (1088, 1360, 1728, 2160, 2592, 3072). We train each model on 500K, 1M, and 2M steps. We always fix the patch size to $14 \times 14$ and the number of attention heads to 16.

## B.3  Grid Sweep

In the grid sweep, we pretrain each architecture on 600M examples. We use the cross-product of:

1. `width`: 416, 512, 608, 768
2. `depth`: 6, 8, 10, 12
3. `MLP Size`: 768, 928, 1088, 1360

Some important considerations to be taken into account include:

- When designing the grid sweep, we made sure that the compute-optimal model selected lies strictly in the *interior* of the grid, not on its boundary. This is because if it lies at the boundary (e.g. its depth is the maximum depth used in the grid), one cannot determine if it is compute-optimal or if increasing that dimension will yield even better models. This can be an iterative process, in which additional grid points are added to the sweep if necessary.

- When identifying the model, we ensured that it is compute-optimal for a good range of compute (not only at some isolated point). Since the model is now compute-optimal for a range of compute budgets, we select as a starting point in our recipe the *least* compute it is optimal for. For example, if a model is compute-optimal for computes ranging from 1 TFLOPs to 2 TFLOPs, we use 1 TFLOPS in our recipe. In other words, we err on the side of caution, giving preference to larger models as we scale up the vision transformer (ViT).

- Generally, the grid sweep should be tightly packed; e.g. with increments of 20% only in each dimension. By contrast, increments in the star sweep should be large in order to identify the scaling exponents reliably.

- Even though the goal in the grid sweep is to identify a "small" architecture that is compute-optimal for a "small" amount of compute, the amount of compute used in the analysis should be large enough for results to be reliable and for power laws to take effect. That is why in our experiments, we use $> 100$M training examples in the grid sweep as opposed, for instance, to using only a few million examples.

## C  Multitask Decoding Setup

Table 6: Multi-task decoding Hyperparameter Settings.

| | |
|---|---|
| Image Resolution | $224 \times 224$ |
| Batch size | 512 |
| Preprocessing | Rescale(-1, 1), ResizeSmall(256), CentralCrop(224) |
| Augmentation | InceptionCrop(224), Left-Right Flip |
| Optimizer | AdaFactor [60] |
| Epochs | 50 |
| Gradient Clipping | 1.0 |
| Label Smoothing | 0.1 |
| Learning Rate | 3e-4 |
| Weight Decay | 1e-4 |
| Schedule | Cosine, 10% Warmup period |
| Vocabulary Size | 32k |
| Encoder Dropout Rate | 0 |
| Decoder Dropout Rate | 0.1 |

Table 6 summarizes the hyperparameter settings for the multitask decoding setup in Section 4.2 and Section 5.3. We always fix the decoder to 2 layers since it generally performs well [8].

# D  LiT Training Setup

Table 7: Locked-image text tuning (LiT) Hyperparameter Settings.

| | |
|---|---|
| Image Resolution | $224 \times 224$ |
| Batch size | 32K |
| Preprocessing | Rescale(-1, 1) |
| Augmentation | None |
| Optimizer | AdaFactor [60] |
| Total Examples | 900M |
| Gradient Clipping | 1.0 |
| Learning Rate | 1e-3 |
| Weight Decay | 1e-4 |
| Schedule | Cosine, 20% Warmup period |
| Vocabulary Size | 32k |
| Bias Init | -10 |
| Temperature Init | 10 |
| Internal Representation | 1,152 |

Table 7 summarizes the hyperparameter settings for the locked-image text turning (LiT) setup, which is used to report zero-shot classification accuracy in Table 4. We use a large batch size of 32K in this setup because it improves the performance of contrastive training [53].

Table 8: ImageNet fine-tuning settings. Settings in the first section vary with resolution, settings in the middle section were explored, and settings in the last section are unexplored good defaults.

| | Full model fine-tuning | | |
|---|---|---|---|
| | 224 px | 384 px | 518 px |
| Learning rate decay | 0.85 | 0.9 | 0.9 |
| Random augment | - | 2,10 | 2,10 |
| Mixup | - | 0.2 | 0.2 |
| Training duration | 50 k steps (20 epochs) | | |
| Learning rate | 0.03 | | |
| Polyak averaging (EMA) | - | | |
| Optimizer | SGD with 0.9 Momentum | | |
| Gradient clipping | 1.0 | | |
| Weight decay | - | | |
| Batch size | 512 | | |
| Learning rate schedule | Cosine with 500 steps linear warmup | | |
| Image crop | `inception_crop` (RandomResize) | | |
| Random flip | Horizontal | | |
| Loss | Sigmoid cross-entropy [6] | | |
| Head init | kernel=0, bias=-6.9 | | |
| Train and minival splits | `train[:98%]` and `train[98%:]` | | |

# E  Transfer to ImageNet-1k

## E.1  Full model fine-tuning

Table 8 lists the settings for the ImageNet-1k fine-tuning results presented in Table 1 in the main paper. The only three settings which differ across resolutions are learningrate decay, random augment and mixup strenghts. We did explore various learningrates, training durations (mostly shorter) as well as Polyak averaging, although the same setting shown in the table appears to be best across the board. Finally, we list various other settings which we did not explore. We simply used good default values from experience.

## E.2  Linear probe on frozen encoder

We take the image representation at the pre-logits, i.e. the 1152-dimensional vector that comes out of the MAP-head and feeds right into the linear classification layer. For each of ViT-L/16, SoViT-400m/14 and ViT-g/14, we perform a grid-search over the following settings, and select the best-performing model on minival (2% of train) to be reported in Table 2: **Augmentation**: `resize(256)|random_crop(224)` vs. `inception_crop(224)`, **learning rate**: 0.001, 0.0003, 0.0001, **epochs**: 1, 3, 10, **weight decay**: 0.0001, None. It should be noted that we keep various other settings to "known good defaults" based on prior explorations with similar models (i.e. plain ViTs). Table 9 summarizes key settings.

Table 9: ImageNet linear probing settings. Settings in the first section were grid-searched for each model, settings in the last section are unexplored good defaults.

| | Linear probe at 224 px | | |
| --- | --- | --- | --- |
| | ViT-L/16 | SoViT-400m/14 | ViT-g/14 |
| Learning rate | 0.001 | 0.0003 | 0.001 |
| Weight decay | 0.0001 | - | - |
| Training duration | 24.7 k steps (10 epochs) | | |
| Image crop | `resize(256)\|random_crop(224)` | | |
| Random augment | - | | |
| Mixup | 0.1 | | |
| Learning rate decay | - | | |
| Polyak averaging (EMA) | - | | |
| Optimizer | SGD with 0.9 Momentum | | |
| Gradient clipping | - | | |
| Batch size | 512 | | |
| Learning rate schedule | Cosine with 10% linear warmup | | |
| Random flip | Horizontal | | |
| Loss | Sigmoid cross-entropy [6] | | |
| Head init | kernel=0, bias=-6.9 | | |
| Train and minival splits | `train[:99%]` and `train[99%:]` | | |

