# OpenReview forum: "Getting ViT in Shape: Scaling Laws for Compute-Optimal Model Design"
_NeurIPS.cc/2023/Conference — NeurIPS 2023 poster_

### Official Review · Reviewer_hmtB · 2023-07-06

**Soundness:** 3 good
**Presentation:** 4 excellent
**Contribution:** 3 good
**Rating:** 6
**Confidence:** 4

**Summary:**

Scaling laws in LLMs have typically been used to derive compute optimal model sizes. In fact one of the initial scaling laws papers in language modeling has indicated that as long as the model size is kept constant, the model shape (corresponding to embedding dim, mlp ratio, number of heads, depth) are not as important for performance. This paper on the contrary discovers that in the case of a Vision Transformer, given an equivalent amount of compute it is indeed feasible to design parameter and inference cost optimal models. These models are competitive with the larger models or outperform them on different tasks. Based on these observations the paper presents qualitative insights for scaling individual shape dimensions of a vision transformer across domains. Further the models derived based on these insights are evaluated on several computer vision tasks.


**Strengths:**

The paper empirically investigates the effect of optimizing shapes of vision transformers to yield competitive models at a fraction of parameter size. Given the parameter and inference cost efficiency this finding is quite impactful towards designing hardware aware vision transformers.
Furthermore, though the empirical investigation is expensive it is at a cost of significantly fewer experiments in comparison to previous scaling laws studies. The paper is very well written and clear in most parts.


**Weaknesses:**

The insights derived in the paper are very important and impactful. However since this is more of an empirical investigation to derive scaling behaviours I find the paper lacks originality. Also the scaling methodology and behaviour would very likely change for different vision applications which also use transformers eg: superresolution, self-supervised learning. Scaling behaviors might also change for different transformer types (swin, deit etc). Deriving scaling laws for every application would mean repeating the analysis, which would incur significant computational cost. Furthermore since the JFT dataset and the code of the paper is not open-sourced the analysis and observations derived are not reproducible. For Zero shot transfer, Linear probes only and imgnet finetuning ViT-G/14 still seems to dominate in most cases, hence I am not very convinced if the representations learned by the SoViT-400m are indeed comparably or more effective than the ones learnt by ViT-G/14.


**Questions:**

1. For given parameter size, does optimizing shape i.e trading-off dimensions while parameter size is held constant help? This was not found to not be helpful in [Kaplan et-al](https://arxiv.org/abs/2001.08361) and it would be interesting to know the observations in this study.
2. How closely are the laws derived tied to the size of the pre-training dataset, would the observations be similar if the laws were studied on imagenet for example?
3. Could the authors report the total TPU-hours/cost of analysis?
4. Minor:  Should scaling “vision transformations” be scaling “vision transformers”? Line 70
5. How does this analysis compare to NAS methods (eg: [AutoFormer ](https://openaccess.thecvf.com/content/ICCV2021/papers/Chen_AutoFormer_Searching_Transformers_for_Visual_Recognition_ICCV_2021_paper.pdf)) which automatically derive optimal shapes? What are the advantages and disadvantages between these?
7. How hyper-parameters are set for different shapes is unclear to me. Could you please clarify this?
8. What do the question marks in Table 1 indicate?
9. ViT-g/14 seems to work better for multitask decoding in table-3. Do the authors have an intuition of why that could be?

If all my concerns and questions are addressed appropriately I am willing to increase my score for the paper.


**Limitations:**

The limitations of the work are not adequately discussed:
1. The compute cost of the analysis?
2. Reproducibility of the experiments?
3. Are there any implicit assumptions which may effect the laws derived empirically in the paper?

I encourage the authors to discuss these limitations in the paper.

---

> ### Author Rebuttal · Authors · 2023-08-09
>
> We thank the reviewer for the detailed and careful review. Please see our response below:
>
> - We disagree that our work lacks originality. We provide an approach based on scaling laws for inferring compute-optimal model shapes, which has not been done in the literature before. In addition, we introduce the functional form in Eq 3, which helps in reducing the number of experiments significantly. We also demonstrate that a combination of a star and a grid sweep is sufficient for inferring compute-optimal shapes, thereby reducing the cost of the analysis further.
>
> - We agree that having different architectures on different domains may require repeating a similar analysis. However, the fact that different domains may result in different optimal shapes is not a limitation of our work per se, since our goal is to discover those optimal shapes, not alter them. We would like to emphasize, however, that the scaling exponents are similar across the two domains we studied. Hence, in terms of order-of-magnitude, the optimal shape is similar in both domains when the model is sufficiently large. We provide evidence to support this in our experiments by evaluating SoViT-400M in multimodal tasks, such as zero-shot classification, captioning, and VQA.
>
> - Regarding open-sourcing the code, we use a publicly available codebase that was removed to preserve anonymity and will be included in the final version of the paper. In the Appendix, we do provide the full training configuration for our experiments and provide a full description of the star and grid sweeps.
>
> - Optimizing the shape while keeping the parameter count fixed indeed helps. We demonstrate this in Figure 6 (leftmost figure). Here, both SoViT-150m and the baseline (denoted B-150m) have the same size, yet SoViT-150m performs better.
>
> - In our analysis, we assume that no overfitting occurs. This is crucial for Eq 2 to be valid, otherwise increasing the compute t can increase the loss at some point. To ensure that there is no overfitting, the size of the pretraining dataset should be quite large. For this reason, we do our analysis on JFT-3B.
>
> - The relation between TPU hours and GFLOPs is reported in Figure 4.
>
> - Thank you for spotting the typo in Line 70. We’ll fix it.
>
> - The same hyperparameters are used for all architectures. These are provided in the Appendix (e.g. see Appendix B.1 for supervised pretraining and Appendix C for multitask decoding).
>
> - We will fix the question marks in Table 1 and replace them with dashes. Thanks for pointing this out.
>
> - The difference between SoViT-400m and ViT-g/14 in Table 4 for multitask decoding is negligible in our opinion; compare it for instance with the difference between SoViT-400m with ViT-L/16 that is similar to it in size.

---

> > ### Author Response · Authors · 2023-08-12
> > **Clarification**
> >
> > Dear reviewer,
> >
> > We thank you again for the insightful feedback and acknowledge the areas where clarification is required.
> >
> > We would like to clarify that in places where we have not detailed how certain feedback will be incorporated, it's mainly because we are still deliberating on the best way to incorporate those suggestions. We are definitely taking all comments into account when revising the paper.
> >
> > This includes, for example, improving the clarity of Figure 1, Figure 3, and Section 3. In addition, we plan to include further details about the CIDEr and log-perplexity metrics, what “equivalent-compute” means, a brief description of the shape dimensions, the meaning of the exponents b and c, a link to the code, and adding further discussions in Section 5.5 to highlight the role of the sequence length. We will address all of these points.
> >
> > Thank you again for the constructive feedback, and for your suggestions to enhance the quality and readability of the paper.

---

### Official Review · Reviewer_s4n2 · 2023-07-06

**Soundness:** 3 good
**Presentation:** 3 good
**Contribution:** 3 good
**Rating:** 7
**Confidence:** 4

**Summary:**

The authors study the recent empirical insight that test performance follows a predictable power-law structure in terms of (optimally-allocated) compute and extend this notion to take into account the “shape” parameters of underlying model such as width, depth etc. They demonstrate that power-law behaviour can indeed be leveraged to design a strategy to explore the shape space, enabling a significant decrease in the amount of computation needed, compared to a naive grid search. The authors focus on the class of vision transformers and show that their discovered shape-optimal models match or outperform larger models (trained with same compute) while at the same time offering more efficient inference.


**Strengths:**

1. The idea to leverage the (predictable) powerlaw behaviour of test performance to determine the ideal shape of a network is very well founded, and it’s very surprising that nothing similar has been done before. The results show that is indeed worthwile to optimize for the shape of vision transformers, as one can apparently achieve similar performance with a significantly smaller model.
2. An important quality of this work which in my opinion goes beyond most previous work on scaling behaviour is that the predictability is actively leveraged to reduce the search space of compute optimal models. By devising the so-called star-sweep strategy, the amount of compute needed is strongly reduced (albeit still very large).
3. The experimental setup is very extensive and the discovered architecture is evaluated on a broad range of tasks, ensuring that its optimality is not just an artifact of optimizing for ImageNet downstream accuracy.


**Weaknesses:**

The definitions of some terms in this paper are sometimes unclear or never really stated. The most prominent and important one is compute **t**. Its definition seems to switch from context to context, which makes it difficult to follow. As far as I understood, compute **t** refers to the total compute, i.e. how many samples/tokens, training epochs and FLOPs per forward pass are needed. In other parts of the text compute seems to be referred as the number of samples solely, e.g. in line 127, compute unbounded seems to mean infinite sample size. Similarly in line 133, how can compute be fixed if the model size can be scaled arbitrarily (and compute is a function of model size)? In general I find it confusing that equation (2) depends both on compute **t** and shape parameter x_k, while **t** iself is actually also dependent on $x_k$, i.e.  $t(x_k)$. Wouldn’t it be cleaner to simply replace compute **t** by the number of samples? I would appreciate if the authors could elaborate on this. I don’t think this has any major implications for the results but a clarification would certainly enhance readability.

Similarly, a quick definition of width, MLP dimension and depth could be helpful for readers less familiar with the details of vision transformers.


**Questions:**

1. How does each shape parameter (i.e. width, MLP dim and depth) contribute to the total number of FLOPs for say, a single forward pass? Is it for instance “cheaper” to make a ViT wider compared to adding another layer?  What about wall-clock time and memory? It would be nice to at least have an approximate understanding of how the FLOP count is affected by these shape dimensions.
2. Something that is maybe clear and I missed it, but were the ViTs that you compare against, i.e. VIT-G/14 and ViT-g/14, considered “compute-optimal” before this work, and thus a “valid baseline”?
3. How does flexifying the architecture rule out that other patch sizes could be shape-optimal? The corresponding section is very short unfortunately (for space reasons I assume) but it would be great if the authors could expand on this.

**Limitations:**

The authors have addressed the limitations of their work.

---

> ### Author Rebuttal · Authors · 2023-08-09
>
> We thank the reviewer for the detailed and careful review. Please see our response below:
>
> - Compute is defined in terms of FLOPs throughout the paper. However, when the architecture is fixed, compute becomes proportional to the number of seen examples. That’s why in Line 127, we refer to infinite data as compute-unbounded.
> There is indeed a dependence between the architecture and its compute as you mention. However, we can always treat them separately by first fixing the architecture and, then, training for the chosen amount of compute (in FLOPs). This is why we can have both $t$ and $x$ in Equation 2. The reason we do not use the sample size is because large models are more sample-efficient (Zhai, et al. 2022) so if the goal is to minimize compute by minimizing the size of the training data, the solution is to (trivially) scale up the size of the model indefinitely.
>
> - We will add a brief description of the shape dimensions in the revised version of the paper.
>
> - The impact of the model shape on compute is captured by the exponent $b$, which are shown in Figure 5. These are automatically taken into account when optimizing the shape for compute in Eq 4. Note that in Eq 4, increasing the value of $b$ would decrease the scaling exponent $s$.
>
> - The baselines ViT-g/14 and ViT-G/14 are not compute-optimal. This is what we demonstrate in this work. We use them because they are widely used in the literature, and hence serve as valid baselines.
>
> - Flexifying does not rule out that other patch sizes could be compute-optimal. Our intent is merely to demonstrate that SoViT-400M continues to perform quite well for other patch sizes when it is flexified.

---

> > ### Author Response · Authors · 2023-08-12
> > **Clarification**
> >
> > Dear reviewer,
> >
> > We thank you again for the insightful feedback and acknowledge the areas where clarification is required.
> >
> > We would like to clarify that in places where we have not detailed how certain feedback will be incorporated, it's mainly because we are still deliberating on the best way to incorporate those suggestions. We are definitely taking all comments into account when revising the paper.
> >
> > This includes, for example, improving the clarity of Figure 1, Figure 3, and Section 3. In addition, we plan to include further details about the CIDEr and log-perplexity metrics, what “equivalent-compute” means, a brief description of the shape dimensions, the meaning of the exponents b and c, a link to the code, and adding further discussions in Section 5.5 to highlight the role of the sequence length. We will address all of these points.
> >
> > Thank you again for the constructive feedback, and for your suggestions to enhance the quality and readability of the paper.

---

> > ### Comment · Reviewer_s4n2 · 2023-08-15
> >
> > I thank the authors for their explanations. I only have a remaining question regarding the definition of compute:
> >
> > **Compute:** Maybe there was a misunderstanding, I'm not suggesting to equate compute $t$ with sample size $N$, but rather have a formula in the Chinchilla style, i.e. $f_k(x_k, N) \propto A_kx_k^{-\alpha_k} + B_kN^{-\beta_k}$ while compute $t \propto N g(x_1, \dots, x_K)$ is fixed where $g(x_1, \dots, x_K)$ determines the number of FLOPs for shape configuration $(x_1, \dots, x_K)$. Of course one could apply your refined formulation to the above law. Or does such an approach still not work? I might still be missing something.

---

> > > ### Author Response · Authors · 2023-08-17
> > > **Response**
> > >
> > > Thank you for the clarification and the suggestion.
> > >
> > > The Chinchilla style formula is a special case of the one we use in (2). In particular, if one has $f_k(x_k, N)\propto A_kx_k^{-\alpha_k} + B_kN^{-\beta_k}+\varepsilon_k$, then Equation 2 would also hold; e.g. by setting $\gamma=0$ and writing: $t \propto x_k^{b/c} N$  (since compute $t$ is directly proportional to the data size $N$ and $t=0$ whenever $x_k=0$). In our experiments, we found this to be a good approximation indeed. In particular, setting $\gamma=0$ results in values of $b$ and $c$ in which the relation $t \propto x_k^{b/c} N$ holds approximately. The scaling exponent $s$ remains relatively unchanged in either case (e.g. in depth, it becomes 0.43 instead of 0.45. The reason it does not change much is because $\gamma\ll 1$ in the first place, as would be expected from Equation 3 and the way we construct the star sweep.
> > >
> > > Thank you for bringing this up. We will add a discussion about it to the paper.

---

### Official Review · Reviewer_DDss · 2023-07-06

**Soundness:** 4 excellent
**Presentation:** 3 good
**Contribution:** 4 excellent
**Rating:** 6
**Confidence:** 5

**Summary:**

This paper proposes a novel and empirical take on the design of large vision transformers (ViTs), in the continuity of a previous paper aiming at optimizing the training of transformers.
Whereas the previous paper was aiming to optimize a single parameter (optimal model size) given a fixed training budget, this paper goes one step further and attempts to discover the optimal ViT architecture (that is 3 parameters: token dimension, depth, MLP size) given a fixed training budget.
The paper empirically solves this question via numerous experiments, while answering related questions and distilling interesting insights on the way.

**Strengths:**

### Writing
- The introduction is well written and clear
- quite complex and unclear at times

### Method
- The optimization method is novel and an improvement w.r.t. the "optimal model size" paper, as it can deal with multiple hyper-parameters while requiring much less experiments.
- The insights about the choice of the joint functional form (eq. (2)) are interesting and well-grounded
- Definitely a valuable paper for the community, even though some aspects could be improved

### Experiments
- multiple experiments on several benchmarks support the initial claim that a smaller-but-optimal architecture achieves as well as vanilla SotA architectures
- Experimental findings are useful and valuable (Section 4.1)
- it is nice to see that other tasks than image classification are being experimented with, and that the findings hold for a wide variety of downstream tasks

**Weaknesses:**

### Exposition
- line 57: "Figure 1: The MLP dimension is scaled faster than depth, which in turn is scaled faster than width." --> I do not find this to be clear, looking at Figure 1

### Method
- The scaling parameters defined in eq (4) seems central in the analysis. Unfortunately, I'm not sure to understand exactly what it is and where it comes from. In particular, I don't understand why it should be invariant to the choice of the shape dimension (see Figure 5)
- About the difference between small and large models
  - it is claimed that "in small models, an optimal shape in one domain is not necessarily optimal in others." (Figure 3)
  - How are "small" and "large" models exactly defined? This seems like a very convenient subjective definition.
  - Why does the model size matter so much when it comes to the optimal architecture across application domains?

### Experiments
- Figure 6:
  - "while keeping compute fixed." --> how exactly? Does this mean other model hyper-parameters are decreased to keep the same model complexity? Or does this mean the training is short/longer?
  - the supposedly optimal model is clearly not optimal, since increasing the depth or MLP size results in (slightly) better performance. Since there experiments are on a relatively "small" model, according to the paper, does this mean the claims do not hold anymore? (see above)


**Questions:**

I'm ready to upgrade my rating if the authors clarify some of the points mentioned above.

**Limitations:**

yes

---

> ### Author Rebuttal · Authors · 2023-08-09
>
> We thank the reviewer for the detailed and careful review. Please see our response below:
>
> - In Figure 1, we use different axes for different dimensions. This may make it difficult to see how the MLP dimension is scaled faster than the others. Note, for example, that going from 1T to 100T GFLOPs corresponds to an increase in MLP by around a factor of $\times 3$. For depth, on the other hand, the increase is around a factor of $\times 2$. Generally, the scaling exponent for the MLP dimension is $\approx 0.6$, which is larger than depth or width.
>
> - Equation 4 is obtained by setting the derivative in (3) to zero and solving for $x$. The scaling exponents need not be invariant to the choice of the dimension as you mention. However, based on prior theoretical works that explain scaling laws via a space partitioning argument (Bahri, et al. 2021; Hutter, 2021; Sharma and Kaplan, 2022), we would expect the exponent $c$ to be roughly similar across all dimensions, which seems to be indeed the case in Figure 5. So, only $c$ is expected to be invariant.
>
> - Regarding the model size, the terminology “small” and “large” is meant to convey in simpler terms the following results. In Figure 3, we observe that the compute-optimal model for classification highlighted in blue is not compute-optimal for image-to-text tasks as shown in the rightmost figure. For this reason, a compute-optimal shape in one domain may not be optimal in others. However, we also show that the scaling exponents are similar. Hence, in terms of order-of-magnitude, the optimal shape is similar in both domains when the model is sufficiently large. We do provide evidence to support this in our experiments by evaluating SoViT-400M in multimodal tasks, such as zero-shot classification, captioning, and VQA.
>
> - We keep compute fixed by changing the training duration, making it longer for smaller models and shorter for larger models, such that the total FLOPs is the same.
>
> - Regarding the optimality of the shape and Figure 6, we clarify this in Lines 115-116. Due to modeling assumptions, approximations, and the finite possible number of experiments, we can only approximate the optimal shape. In Figure 6, we see that deviating from the predicted optimal width, for example, does degrade performance. For other dimensions, the performance does not change significantly when they are increased but it does degrade when they are decreased. We do believe that this highlights our argument that one can identify a near-optimal shape using the recipe we propose.

---

> > ### Comment · Reviewer_DDss · 2023-08-11
> > **acknolegment**
> >
> > I have read the rebuttal, and I must say I am relatively disappointed by the lack of willingness from the authors to improve their paper based on my and other reviewer suggestions. Overall, I see no promise to improve the manuscript except for fixing typos and adding references.
> >
> > I know what is a plot and an axis, and I know how to read them, thank you. My role as a reviewer is not just to accept or reject a paper, it is also to help improve the paper quality and readability, for the sake of readers. Arguably, Figure 1 is a good example of my point: it is cited in the paper line 56, followed by the explanation "The MLP dimension is scaled faster than depth, which in turn is scaled faster than width", but this does not show at all at first sight when someone looks at Figure 1! And not on 2nd sight either. It actually requires some calculations based on the axes' tick labels to realize than, indeed, MLP dimension is scaled faster than depth, etc. The figure is completely counter-intuitive in that regard. Is it too much to ask for improving it?
> >
> > Same comments for the scaling exponents. Section 3 is really technical, and it would't hurt to explain things a bit more (as pointed out by other reviewers too).
> >
> > Same comments for other things the reviewer found difficult, unclear or counter-intuitive.
> >
> > Also, I am not satisfied by the answer regarding the difference between large and small models. I know that asymptotically, shape will not matter when the model grow sufficiently large. What would be useful is to be able to characterize when this happens, because so far the paper does not answer this question at all. By the way, Figure 3 is, again, not super clear nor intuitive (and what does all the gray circle exactly denote?)

---

> > > ### Author Response · Authors · 2023-08-12
> > > **Clarification**
> > >
> > > We appreciate your prompt and insightful feedback and acknowledge the areas where clarification is required that are highlighted by the reviewers.
> > >
> > > In places where we have not detailed how certain feedback will be incorporated, it's mainly because we are still deliberating on the best way to incorporate those suggestions. We are definitely taking all comments into account when revising the paper.
> > >
> > > In reference to Figure 1, in particular, we plan to make it easier to see the rate of growth of each dimension as you suggested. Possible approaches might include adding a second right-axis to display the percentage increase from a fixed reference point (such as 1T tokens), or adding labels within the plot to indicate specific milestones, like when a dimension is doubled or tripled.
> > >
> > > The same holds for Section 3 and the other places suggested by the reviewers, such as Figure 3, details about the CIDEr and log-perplexity metrics, what “equivalent-compute” means, a brief description of the shape dimensions, clarifying the meaning of the exponents b and c, a link to the code, and adding further discussions in Section 5.5 to highlight the role of sequence length. We will address all of these points.
> > >
> > > Regarding the model size, we will rephrase that statement to say that the optimal shapes can be initially different but they converge as the model size increases, without referring explicitly to the terms “small” or “large” since that can be subjective as you suggested.
> > >
> > > Thank you again for the constructive feedback, and for your suggestions to enhance the quality and readability of the paper.

---

> > > > ### Comment · Reviewer_DDss · 2023-08-15
> > > > **Re: Clarification**
> > > >
> > > > >  it's mainly because we are still deliberating on the best way to incorporate those suggestions. We are definitely taking all comments into account when revising the paper.
> > > >
> > > > I'm relieved to hear that. About Figure 1, that sounds like a good idea. If I may suggest, isn't it even better to simply make sure the log-scale of the  y-axis is the same in all 3 plots? Sure, that would squeeze a bit some axes, but at least it would perfectly convey the key finding of the paper. Well, this is just an idea, it may render poorly.
> > > >
> > > > > Section 3
> > > >
> > > > Ok great.
> > > >
> > > > > Regarding the model size
> > > >
> > > > I would even suggest to clarify why optimal shapes should converge when the model size grows. I believe this is not completely obvious.

---

> > > > > ### Author Response · Authors · 2023-08-17
> > > > > **Thank you**
> > > > >
> > > > > Thank you for the continued engagement and the great suggestions. We will add a precise statement about the convergence of the optimal shapes to the paper.
> > > > >
> > > > > Unfortunately, using the same log-scale in Figure 1 for all dimensions will make some of the plots unreadable, particularly depth.

---

### Official Review · Reviewer_T7K3 · 2023-07-08

**Soundness:** 3 good
**Presentation:** 3 good
**Contribution:** 3 good
**Rating:** 7
**Confidence:** 4

**Summary:**

This paper introduces an efficient approach to investigate the scaling laws for compute-optimal model shapes, such as model width and depth. It proposes a shape-optimized vision transformer called SoViT.
A comprehensive evaluation across various tasks highlights the effectiveness of the proposed architecture. SoViT-400m/14 achieves 90.3% fine-tuning accuracy on ILSRCV2012, surpassing  ViT-g/14 with a larger model size. This study makes a valuable contribution to the design of vision transformers and is expected to have a certain degree of impact in this era.

**Strengths:**

1. The paper is well-written, with well-defined formulas and sufficient supporting materials.
2. The three shape parameters are analyzed well. The paper proposes star sweep and grid sweep strategies to investigate the scaling laws avoiding the expensive search cost.
3. Extensive experiments are implemented to validate the method, including image classification,  multitask decoding, and segmentation tasks.


**Weaknesses:**

1. Figure 3. is challenging to understand, and it may benefit from more elaborate annotations and explanations to clarify the significance of certain data points within the figure.
2. For the experiments, SoViT-400m/14 can surpass ViT-g/14 for the image classification task. However, SoViT-400m/14 does not show a significant advantage over ViT-g/14 and ViT-L/16 in other tasks, such as the OCR and VQA. The detail of metrics such as Log-PPL and CIDEr in Table 4. should be explained in the paper.
3. A few typos in the paper, such as x_{k} in A.1 Quasiconvexity Proof.


**Questions:**

1. Figure 6. indicates that deviating from the optimal depth/MLP configuration does not lead to performance degradation. Furthermore, the evidence supporting the superiority of the optimal shape is limited, and it seems that only three experiments (33%<, 33%>, 200%) were conducted, as shown in the Figure.
2. The compute-optimal model shape is different among tasks. How to search for an optimal model which can be applied to various downstream tasks?

**Limitations:**

As above.

---

> ### Author Rebuttal · Authors · 2023-08-09
>
> We thank the reviewer for the detailed and careful review. Please see our response below:
>
> - In Figure 3, each dot corresponds to a model architecture pretrained on 600M examples and evaluated on one downstream metric.
> The metrics  from left to right are: 5-shot, 10-shot, and 25-shot (all in ImageNet). In the rightmost figure, the metric is an average log-perplexity score across a mixture of four tasks including VQA and captioning, as we describe in Section 4.2.
> In the first three figures, the downstream metrics are for classification and we observe that the compute-optimal model highlighted in blue is compute-optimal in all three cases because it lies in the efficient frontiers in all three cases. But, it is not compute-optimal for image-to-text tasks as shown in the rightmost figure. For this reason, a compute-optimal shape in one domain may not be optimal in others. We will clarify this in the paper.
>
> - In Table 4, we show that SoViT-400M is comparable to ViT-g/14 in multimodal tasks. As you mention, it does not perform strictly better but performing equally well is itself a significant gain, because SoViT-400M is much smaller and less costly (GFLOPs, Images/Core/s) than ViT-g/14, as shown in Figure 2.
>
> - We did not explain log-perplexity and CIDEr because they are standard metrics in the literature (e.g. [1] and Section 9.3.2 in [2]). We will include references for their definitions in the revised version of the paper.
>
> - Thanks for spotting the typo in A.1. We’ll fix it.
>
> - Regarding the optimality of the shape and Figure 6, we clarify this in Lines 115-116. Due to modeling assumptions, approximations, and the finite possible number of experiments, we can only approximate the optimal shape. In Figure 6, we see that deviating from the predicted optimal width, for example, does degrade performance. For other dimensions, the performance does not change significantly when they are increased but it does degrade when they are decreased. We do believe that this highlights our argument that one can identify a near-optimal shape using the recipe we propose.
>
> - The reason we only report <33%, >33%, and >200% in Figure 6 is because we do observe a drop in all cases when using <33% so it is a sufficient demonstration. On the other hand, increasing the dimension by >33% or >200% both give similar results, so we expect similar results when using, for example, >100%.
>
> - It is true that the compute-optimal shape is different across domains when the model is small. The scaling exponents, however, are the same in both domains: (1) image classification and (2) image-to-text, as discussed in Section 4.2 . This means that in terms of order-of-magnitude, the optimal shape is similar in both domains when the model is sufficiently large. We do provide evidence to support this in our experiments by evaluating SoViT-400M in multimodal tasks, such as zero-shot classification, captioning, and VQA.
>
> [1] Vedantam, R, et al. "Cider: Consensus-based image description evaluation." CVPR, 2015.
>
> [2] Zhang, A., Lipton, Z. C., Li, M., & Smola, A. J. Dive into deep learning. 2021.

---

> > ### Author Response · Authors · 2023-08-12
> > **Clarification**
> >
> > Dear reviewer,
> >
> > We thank you again for the insightful feedback and we acknowledge the areas where clarification is required.
> >
> > We would like to clarify that in places where we have not detailed how certain feedback will be incorporated, it's mainly because we are still deliberating on the best way to incorporate those suggestions. We are definitely taking all comments into account when revising the paper.
> >
> > This includes, for example, improving the clarity of Figure 1, Figure 3, and Section 3. In addition, we plan to include further details about the CIDEr and log-perplexity metrics, what “equivalent-compute” means, a brief description of the shape dimensions, the meaning of the exponents b and c, a link to the code, and adding further discussions in Section 5.5 to highlight the role of the sequence length. We will address all of these points.
> >
> > Thank you again for the constructive feedback, and for your suggestions to enhance the quality and readability of the paper.

---

### Decision · Program_Chairs · 2023-09-21

**Decision:**

Accept (poster)

**Comment:**

This paper explores scaling laws and optimization strategies for ViT shapes / architecture hyper-parameters, introducing a new shape optimized variant of ViTs, evaluated on multiple downstream tasks. The paper received all positive reviews from 4 expert reviewers, who appreciated a well written paper, extensive experiments and analyses,

Minor weaknesses raised were on clarity, in particular some Figures, and metrics. The paper was extensively discussed between reviewers and authors, which led to improvements of the presentation of an already initially well received paper, thus a positive outcome of the peer reviewing process.

The AC recommends acceptance.